# The feasibility, acceptability, and preliminary impact of real-time monitors and SMS on tuberculosis medication adherence in southwestern Uganda: Findings from a mixed methods pilot randomized controlled trial

Angella Musiimenta[1,2]*, Wilson Tumuhimbise[1,2], Esther C. Atukunda[1], Aaron T. Mugaba[1,2], Nicholas Musinguzi[1], Conrad Muzoora[1], David Bangsberg[3], J. Lucian Davis[4,5,6], Jessica E. Haberer[7,8]

1 Mbarara University of Science and Technology, Mbarara, Uganda, 2 Angels Compassion Research and Development Initiative, Mbarara, Uganda, 3 Oregon Health & Science University-Portland State University School of Public Health, Portland, Oregon, United States of America, 4 Uganda Tuberculosis Implementation Research Consortium, Makerere University, Kampala, Uganda, 5 Department of Epidemiology of Microbial Diseases, Yale School of Public Health, New Haven, CT, United States of America, 6 Pulmonary Critical Care and Sleep Medicine Section, Yale School of Medicine, New Haven, CT, United States of America, 7 Harvard Medical School, Boston, MA, United States of America, 8 Massachusetts General Hospital Center for Global Health, Boston, MA, United States of America

* amusiimenta@must.ac.ug

## Abstract

We conducted a pilot randomized controlled trial among patients initiating treatment for drug-sensitive tuberculosis (TB). Participants received real-time electronic adherence monitors and were randomized (1:1:1) to: (i) daily SMS (reminders to TB patients and notifications to social supporters sent daily for 3 months, then triggered by late or missed dosing for 3 months); (ii) weekly SMS (reminders to TB patients and notifications to social supporters sent weekly for 3 months, then triggered by late or missed dosing for 3 months); or (iii) control (no SMS). Feasibility was mainly verified by the technical function of the intervention at Month 6. The primary outcome was percent adherence as ascertained by the real time monitor. Quantitative feasibility/acceptability data were summarized descriptively. Percentage adherence and adherence patterns were assessed and compared by linear regression models. Qualitative acceptability data was collected through interviews and analyzed using content analysis. Among 63 participants, the median age was 35 years, 75% had no regular income, and 84% were living with HIV. Feasibility was demonstrated as most of the daily [1913/2395 (80%)] and weekly [631/872 (72%)] SMS reminders to TB patients were sent successfully. Also, most of the daily [1577/2395 (66%)] and weekly [740/872 (85%)] SMS notifications to social supporters and adherence data (96%) were sent successfully. Challenges included TB status disclosure, and financial constraints. All patients perceived the intervention to be useful in reminding and motivating them to take medication. Median adherence (IQR) in the daily SMS, weekly SMS, and control arms was 96.1% (84.8, 98.0), 92.5% (80.6, 96.3), and 92.2% (56.3, 97.8), respectively; however, differences between the

**Data Availability Statement:** Data from this study contain sensitive patient information. Also, sharing this data breach the compliance with the protocol approved our research ethics board. However, de-identified individual data that supports the results will be shared following publication provided the investigator who proposes to use the data has approval from an Institutional Review Board (IRB), Independent Ethics Committee (IEC), or Research Ethics Committee (REC), as applicable, and executes a data use/sharing agreement with Mbarara University of Science and Technology (MUST). Researchers may apply for data access by contacting the corresponding author or the MUST Research Ethics Committee at sec@rec.must.ac.ug.

**Funding:** Funding: This work was supported by the Fogarty International Center of the National Institutes of Health (K43TW010388 to AM). The contents of this article are solely the responsibility of the authors and do not necessarily represent the official views of the Fogarty International Center of the National Institutes of Health. The funders had no role in study design, data collection and analysis, decision to publish, or preparation of the manuscript. Authors AM, AMT and WT received salaries from the Fogarty International Center of the National Institutes of Health (K43TW010388 to AM).

**Competing interests:** The authors have declared that no competing interests exist.

intervention and control arms were not statistically significant. Real-time monitoring linked to SMS was feasible and acceptable and may have improved TB medication adherence. Larger studies are needed to further assess impact on adherence and clinical outcomes.

**Trial registration**. ClinicalTrials.gov registration number: NCT03800888. https://ichgcp.net/clinical-trials-registry/NCT03800888.

## Introduction

Worldwide, nearly 10 million people develop tuberculosis (TB) and nearly 2 million people die from TB annually [1]. Low income countries account for more than 90% of TB cases and deaths [2]. Uganda faces a high burden of TB and is listed among the 30 countries with the highest rates of TB and TB/HIV co-infection [2], with an annual TB incidence of 196 (95% CI 117–296) per 100,000 overall and 65 (95% CI 39–98) per 100,000 people living with HIV [3]. In Uganda, an estimated 223 people acquire TB daily and 30 deaths due to TB occur [3]. Importantly, TB is the number one cause of death among individuals living with HIV [4]. While TB treatment is freely available in Uganda, treatment adherence challenges remain [5]. Causes of non-adherence to TB medication include depression and alcohol [6], lack of transport to the clinics to pick up drugs, [5] and forgetfulness [7]. Non-adherence to TB medication contributes to disease transmission, drug resistance, and treatment failure [8–10].

The World Health Organization (WHO) has long recommended directly observed therapy (DOT), which requires taking medication under the supervision of a health worker or trained treatment supporter [11]. However, the implementation of DOT has been limited in many settings for several reasons; DOT requires a significant time commitment from people living with TB, healthcare workers, and treatment supporters and places financial burdens on TB patients when they travel to the clinic for DOT administration [12,13]. DOT may also infringe on TB patients' autonomy [14]. Moreover, for preventing disease relapse and acquired drug resistance, DOT may not perform better than self-administered therapy [15]. Alternative, novel methods are needed for effective management of TB medication adherence especially in settings where DOT' implementation is challenging and/or undesirable. Recognizing this need, the WHO now recommends using digital adherence technologies to support TB medication adherence as part of its End TB Strategy [16]. Examples of such technologies include SMS reminders and digital medication monitors (i.e., devices that send signals when opened as a proxy for taking medication). Importantly, real-time adherence monitors can enable the provision of timely interventions to address non-adherence before TB patients experience the consequences of non-adherence. Real-time adherence monitoring linked with SMS reminders has been shown to be feasible and acceptable [17] and to improve adherence to antiretroviral therapy (ART) [18]. However, few quality studies have investigated these technologies for TB medication adherence, especially in low resource settings, and the modest prevailing studies report mixed results [19–23]. In addition, the role for SMS in activating social support among TB patients has not been well studied, and research is needed on the integration of supportive SMS notifications to real-time monitoring of TB medication adherence.

We therefore conducted a pilot randomized controlled trial (RCT) investigating the use of: i) real-time adherence monitors, ii) SMS as adherence reminders to TB patients, and iii) SMS notifications to their social supporters (i.e., friends or family who have helped them with medication or other needs previously) to support TB medication adherence in southwestern Uganda. We previously published formative qualitative findings from the current study

demonstrating that real-time adherence monitors and SMS can potentially enhance TB medication adherence by reminding them to take TB medication, and giving them opportunities to demonstrate their commitment to medication adherence through monitoring [24]. Participants also reported that SMS notifications to social supporters can motivate medication adherence by creating a personal sense of commitment to adhere to medication in response to the helping hand extended by social supporters [5]. The current paper reports on the feasibility, acceptability and the preliminary impact of this intervention composed of a real-time device that monitors how TB patients take their medication, SMS reminders to patients and SMS notifications to their social supporters.

## Materials and methods

### Ethics statement

All participants provided signed informed consent before study participation. The institutional review committees of Mbarara University of Science and Technology (Protocol number: 16/10–16), and the Uganda National Council for Science and Technology (Protocol number: HS 2189).

### Study design and setting

This study reports the quantitative and qualitative results from a pilot RCT among people with drug-sensitive TB recruited from the TB Clinic within Mbarara Regional Referral Hospital (MRRH) in southwestern Uganda. The TB clinic at MRRH provides care to an estimated 600 TB patients annually. At the TB Clinic, newly diagnosed TB patients receive free TB medication and are counseled about the benefits of TB medication. The DOT approach is not employed for monitoring medication adherence at MRRH due to the costs for both TB patients and within the healthcare system. Instead, TB patients self-administer therapy with isoniazid, rifampin, pyrazinamide, and ethambutol for two months (2RHZE; intensive phase), followed by isoniazid plus rifampin for four months (4HR; continuation phase). The clinic verbally encourages patients to adhere to their medication. The TB treatment regimens use fixed-dose combination tablets TB patients return to the TB Clinic every two weeks for the first two months, including for a sputum examination to ensure conversion to negative results prior to the continuation phase. Those with positive test result receive GeneXpert to exclude rifampicin resistance. Treatment can be extended up to a full year to account for missed medication refills or doses.

### Selection of study participants

Between May 2019 and June 2020, we randomly recruited TB patients if they met the following inclusion criteria: a) being age 18 years and older, b) being newly diagnosed with TB per clinic records (within a month of initiating treatment), b) initiating drug-sensitive TB treatment at the MRRH TB Clinic, c) living in Mbarara District (within 20 km of MRRH), d) knowing how to use SMS, e) owning a cellphone for personal use, and f) having reliable cellular network at home. Before enrollment in the study, cellular reception was assessed by the research assistants who escorted participants in their homes to check the signal of the networks supported by the technology used in this study either MTN or Airtel. The exclusion criteria were an unwillingness or inability to have cellular reception confirmed at home and/or to give consent to be in the study. Inclusion criteria for social supporters were a) being age 18 years and older, b) living in the Mbarara District, c) having an ongoing relationship with the TB patients(e.g., friend or family member), d) owning a cell phone for personal use, and e) knowing how to use SMS. The only exclusion criterion was being unwilling or unable to give consent.

## The intervention technology

The intervention was composed of a real-time device that monitors how TB patients take their medication as described elsewhere [17,24], the real-time adherence monitor (Wisepill Technologies, Cape Town, South Africa) is a medication container that holds up to 28 tablets of RHZE. The real-time adherence monitor has an internal modem and subscriber identity module (SIM) card enable the device to send a real-time mobile signal to a secure web server (hosted in South Africa) by General Packet Radio Service. When opened to take pills, the device records a date-and-time stamp; receipt of this signal is interpreted as a proxy for taking medication. The device stores records of openings in the event of inadequate mobile network coverage and sends them when the network becomes available. The monitor additionally sends a daily "heartbeat" to confirm device functionality. The monitor can be charged using electricity or a solar device; its battery life is typically six months. Additionally, the intervention involved SMS reminders (scheduled and/or triggered by missed or delayed dosing) to TB patients to help them take their medications, as well as SMS notifications to their social supporters to provide the TB patients with assistance if possible. The content of SMS reminders and notifications were determined by each participant. The default message for both TB patients and social supporters was "*This is your reminder.*" SMS were sent from Wisepill technologies at no cost to participants at their specific time of taking medication.

## Outcome measures

The primary outcome was percent adherence ascertained by the real time monitor. Secondary outcomes were feasibility which was verified by the technical function of the intervention including the number of SMS sent as planned, number of SMS not sent as planned, number of SMS sent later due to a delay or absence of the corresponding signal, data loss by the real-time monitor, monitor malfunctioning, taking medication from other sources, and opening the device without taking medication. Acceptability of the intervention was verified by the participant's preference of SMS type and frequency, ease of use of the real-time monitor, possibility of unintended TB status disclosure.

## Study procedures

TB patients were followed for six months. A random number generator (random allocation was generated by the study coordinator) and the CONSORT diagram and the CONSORT checklist (S1 Checklist) (Fig 1 below) were used to determine study arm assignments by the study team. After screening and consenting, all participants received a real-time adherence monitoring device and were randomized 1:1:1 as follows:

1. **Daily SMS + real-time adherence monitor -> triggered SMS** ('daily SMS arm'): Daily SMS reminders and notifications were sent to the TB patients depending on their specific time of medication taking and social supporter participants, respectively, for the first three months. For the next three months, SMS reminders and notifications were sent only if the monitor was not opened within an hour of the expected time.

2. **Weekly SMS + real-time adherence monitor -> triggered SMS** (weekly SMS arm): Weekly SMS reminders and notifications were sent for the first three months to TB patients and social supporter participants, respectively. For the next three months, SMS reminders and notifications were sent only if the monitor was not opened within an hour of the expected time.

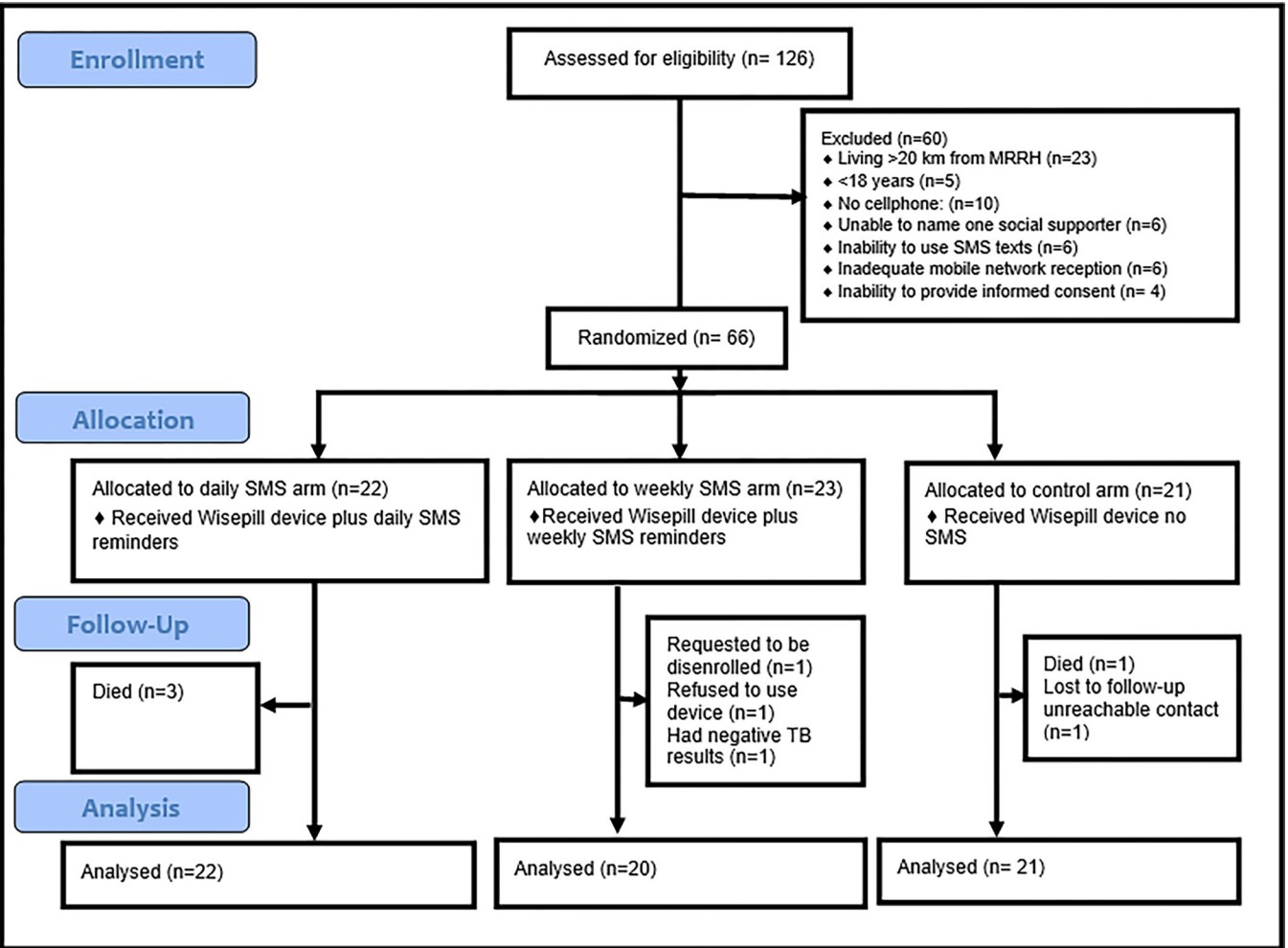

**Fig 1. CONSORT diagram to illustrate participant flow in the study.**

3. **Real-time adherence monitoring only** (control arm): No SMS were sent to TB patients or social supporter participants.

Social supporters were identified by study participants at enrollment and were enrolled into the study at the same time with the patients TB patients. Social supporters were generally encouraged to provide support (e.g. medication reminders, counseling, or transport to the clinic) necessary for enabling study participants take their medication well, if possible.

To manage potential strain on relationships and harmonize expectations [25], the research assistants (WT and ATM) provided orientation to TB patients and social supporters at recruitment in which we clarified their roles and highlighted the potential negative effect of non-responsiveness (not taking medication as expected) on relationships.

## Data collection

Study participants completed baseline questionnaires (S1 Text) regarding socio-demographics, health, depression [26], food insecurity [27], alcohol use [28], social support [29], TB stigma [30], and socioeconomic status [30]. Adherence data was comprised of signals sent to the study server after opening the real-time adherence monitor. Feasibility data was obtained by

tracking SMS sent and received, and monitor and battery functionality. Participant opinion on feasibility and acceptability data was also captured through the study exit pilot-tested questionnaires and qualitative interviews. The development of the questionnaires and interview guide was informed by the Unified Theory of Acceptance and Use of Technology (UTAUT) model [31]. At study exit (month 6), authors WT and ATM (male study research assistants trained in qualitative research, research ethics and Data management with a PhD and MBA respectively) conducted semi-structured, face to face in-depth interviews with TB patients in a private space at a research office near the MRRH until reaching thematic saturation at the 30th participant. Prior to data collection, the researchers had interacted with the participants at enrollment. Each interview lasted between 30 and 60 minutes and was conducted in the local language (Runyankole), digitally recorded, transcribed, and translated to English. The interviewers elicited participants' experiences of using each component of technologies (real-time monitor, SMS reminders for TB patients, and SMS notifications for social supporters) including benefits and challenges related to the technologies. Following each interview, author AM (with support from JEH and JLD) reviewed transcripts for quality, clarity, and detail.

## Analysis

Feasibility and acceptability data were summarized descriptively. In the primary dataset for assessment of adherence, adherence was computed for each participant as the total number of device openings divided by the total number of study follow-up days, excluding days when a participant was considered not to have used the device (e.g., when the device was non-functional as evidenced by no device heartbeat). Adherence data for TB patients who died was included up until death; data were also excluded at loss to follow-up (at the onset of missing data), which was defined as having no contact or adherence data for two months per protocol analysis. Participants were excluded if they voluntarily withdrew from the study or declined to use the adherence monitor. SMS data for TB patients was determined by the Wisepill device in regards to SMS reminders that were sent as planned, those not sent or delayed due to technical challenges like poor network and those sent due to a delay or absence of the corresponding signal. In a secondary analysis, days between loss to follow-up and study completion were considered non-adherence. To assess the effect of each intervention on adherence compared to the control, we used linear regression models with robust standard errors, given the pilot nature of the RCT and the potential for residual confounding in a small sample size. First, we built univariable models considering all potential confounders, which included age, gender, marital status, type of residence, disclosure of TB status to anyone other than their healthcare provider, severe food insecurity, alcohol use, stigma, social support, depression, HIV status, social economic status, study phase (less or equal to 2 months of treatment vs more than 2 months of treatment) and level of education [32] and controlled for the study arm. In the multivariable model, we considered those confounders with a univariable p value <0.2. Analysis was conducted in Stata version 13. In both primary and secondary analysis, the statistician was not blinded to allocation assignments since this was an open label trial.

Qualitative data was analyzed using content analysis [33] to generate categories summarizing TB patients' experiences of using the technologies, focusing on perceived usefulness and challenging issues counting for effort, social influences and facilitators. AM assembled a codebook from the identified categories, using an iterative process, which included developing codes, writing operational definitions, and selecting illustrative quotes. AM then applied codes using NVIVO 11. Qualitative findings are reported according to COREQ guidelines (S1 Checklist) [34]. The pilot trial was reported according to the Consolidated Standards of Reporting Trials (CONSORT) reporting guideline [35].

## Results

### Participant characteristics

Of 126 TB patients screened, 66 (52%) were enrolled in the study between January 2019 and December 2019. TB patients participants were randomized to three arms: daily SMS arm (N = 22), weekly SMS arm (N = 23) and the control arm (N = 21). Reasons for excluding participants are shown in Fig 1. All 45 social supporters identified by the TB patients in the two SMS arms were enrolled in the study.

Overall, the primary analysis considered 22 TB patients in the daily SMS arm, 20 TB patients in the weekly SMS arm, and 21 TB patients in the control arm, totaling to 63 TB patients. However, due to the three patients who died from the daily SMS arm and one who died and the other one who was lost to follow up (due to unreachable contact in the control arm), we only had 58 TB patients at study exit; 19 TB patients in the daily SMS arm, 20 in the weekly SMS arm, and 19 in the control arm. Of the 45 social supporters, three were disenrolled all from the weekly SMS arm after 3.5, 6.5, and 12.2 weeks respectively and their data was excluded for analysis because of disenrollment of their corresponding TB patients. Data from the three social supporters whose TB patients died (from the daily SMS arm) before completing the study were included for analysis. Overall, a total of 39 social supporters completed the study. As indicated in Table 1 below, of 63 TB patients, 32 (59%) were male, 53 (84%) had co-infection with HIV, and 47 (75%) had no regular income.

Of the 42 social supporters, 28 (66%) were female, 16 (38%) had HIV, and 32 (22%) had no regular income.

### Feasibility and acceptability

**SMS reminders and notifications.**   All participants reported that SMS were easy to use (i.e., access and read). As indicated in Table 2 below, most of the daily [1913/2395 (80%)] and weekly SMS reminders [631/872 (72%)] were sent successfully. Also, most of the daily [1577/2395 (66%)] and weekly SMS notifications [740/872 (85%)], were sent successfully. Only 161/2395 (7%) of SMS were sent later due to a delay or absence of the corresponding signal. The sending of these SMS reminders occurred when the electronic adherence monitor had already been opened by the patients to take medication, but there was delay or absence of the corresponding signal due technical issues (such as poor network coverage). Most TB patients in the weekly SMS arm (n = 17; 89%) wished they had received daily SMS reminders to triggered SMS reminders. Responses from the open-ended question on reasons for the stated preferences revealed that daily SMS reminders matched well with daily pill taking and provided encouragement to TB patients daily, while triggered SMS texts were received after the scheduled pill taking time. The majority of the TB patients in the weekly arm (n = 16; 80%) preferred triggered SMS to weekly, reporting that weekly SMS reminders did not match well with daily pill taking. The majority of participants (n = 29; 74%) preferred personalized SMS reminders formulated by themselves (e.g. "*How are you today*") to the default SMS formulated by the study (i.e. "*This is your reminder*").

**SMS notifications to social supporters.**   As indicated in Table 2 above, most of the daily [1577/2395 (66%)] and weekly SMS notifications [740/872 (85%)] were sent successfully. From the survey results, all social supporters (n = 39; 100%) reported providing some assistance to the participants at least once, after receiving SMS notifications. The assistance that involved some form of financial spending (such as providing money for transport to the clinic, buying food) was provided least often (n = 20; 51%) Non-monetary support such as emotional support was the most common support provided (n = 38; 97%). The majority of the social

**Table 1.  Baseline demographic characteristics of participants.**

| Characteristic | TB patients (N = 63) | Social supporters (N = 42) | Stratification of TB patients by arm (n % | | |
| --- | --- | --- | --- | --- | --- |
| | | | Daily SMS (n = 22) | Weekly SMS (n = 20) | Control (n = 21) |
| Median age (IQR) | 35 (28–44) | 35.5 (28–44) | 37.5 (32–54) | 31 (25.5–41) | 37 (30–43) |
| **Gender** | | | | | |
| Male | 37 (59%) | 14 (33%) | 14 (64%) | 13 (65%) | 10 (48%) |
| Female | 26 (41%) | 28 (66%) | 8 (36%) | 7 (35%) | 11 (52%) |
| **Marital status** | | | | | |
| Married | 40 (64%) | 29 (69%) | 16 (73%) | 14 (70%) | 10 (48%) |
| Single | 23 (36%) | 13 (31%) | 6 (27%) | 6 (30%) | 11 (52%) |
| **Type of residence** | | | | | |
| Town | 32 (51%) | 22 (52%) | 8 (36%) | 11 (55%) | 13 (62%) |
| Rural | 31 (49%) | 20 (48%) | 14 (64%) | 9 (45%) | 8 (38%) |
| **Level of education** | | | | | |
| None | 1 (2%) | 2 (4%) | 0 (0%) | 1 (5%) | 0 (0%) |
| P1-P7 | 25 (40%) | 15 (36%) | 12 (55%) | 5 (25%) | 8 (38%) |
| >P7 | 37 (59%) | 25 (60%) | 10 (45%) | 14 (70%) | 13 (62%) |
| **Income (fixed wages/ salary)** | | | | | |
| Yes | 16 (25%) | 10 (28%) | 4 (18%) | 5 (25%) | 7 (33%) |
| No | 47 (75%) | 32 (72%) | 18 (82%) | 15 (75%) | 14 (67%) |
| **Heavy alcohol consumption[a]** | | | | | |
| Yes | 7 (11%) | 4 (10%) | 2 (9%) | 3 (15%) | 2 (9%) |
| No | 56 (89%) | 38 (90%) | 20 (91%) | 17 (85%) | 19 (91%) |
| **TB stigma[b]** | | | | | |
| Median stigma (IQR) | 2 (1–4) | n/a | 2 (1–4) | 2 (1–5) | 2 (1–3) |
| **Enough social support** | | | | | |
| Yes | 49 (78%) | n/a | 17 (77%) | 17 (85%) | 15 (71%) |
| No | 14 (22%) | | 5 (23%) | 3 (15%) | 6 (29%) |
| **Food Insecurity[c]** | | | | | |
| Severe Food Insecure | 33 (52%) | 17 (40%) | 10 (45%) | 11 (55%) | 12 (57%) |
| Food secure | 30 (48%) | 25 (60%) | 12 (55%) | 9 (45%) | 9 (43%) |
| **Probable depression[d]** | | | | | |
| Yes | 11 (18%) | 2 (5%) | 4 (18%) | 3 (18%) | 4 (19%) |
| No | 52 (82%) | 40 (95%) | 18 (82%) | 17 (82%) | 17 (81%) |
| **Asset index[e]** | | | | | |
| Lowest quartile | 17 (27%) | 11 (26%) | 7 (32%) | 5 (26%) | 5 (24%) |
| 25–100% quintiles | 45 (73%) | 31 (74%) | 15 (68%) | 14 (74%) | 16 (76%) |
| **Disclosed TB Status to anyone other than healthcare provider** | | | | | |
| Yes | 52 (83%) | n/a | 18 (82%) | 16 (80%) | 18 (86%) |
| No | 11 (17%) | n/a | 4 (18%) | 4 (20%) | 3 (14%) |
| **HIV status** | | | | | |
| Negative | 10 (16%) | 26 (62%) | 3 (14%) | 3 (15%) | 4 (19%) |
| Positive | 53 (84%) | 16 (38%) | 19 (86%) | 17 (85%) | 17 (81%) |
| **Social support relationship to the study participant** | | | | | |
| Spouse | | 22 (52%) | | | |
| Family member | | 16 (38%) | | | |

(*Continued*)

**Table 1.** (Continued)

| Characteristic | TB patients (N = 63) | Social supporters (N = 42) | Stratification of TB patients by arm (n % | | |
|---|---|---|---|---|---|
| | | | Daily SMS (n = 22) | Weekly SMS (n = 20) | Control (n = 21) |
| Friend | | 4 (10%) | | | |

Notes: [a]Screening for heavy drinking was based on the AUDIT-C (Bush et al., 1998).

[b]TB stigma was measured using the internalized stigma scale (Kalichman et al 2009).

[c]Food insecurity was assessed using the nine item Household Food Insecurity Access Scale (HFIAS) by Coates, Swindale & Bilisnky (2006).

[d]Depression was measured using the measure proposed by Bolton, Wilk, and Ndogoni. (2004).

[e]Asset index was measured using the measure proposed by Filmer and Pritchett (2001).

**Table 2.** Feasibility and acceptability of SMS reminders and notifications.

| **Feasibility and acceptability of SMS reminders to patients** | |
|---|---|
| *N = 39 (i.e. 19 in the daily SMS arm and 20 in the weekly SMS arm who completed the study)* | |
| **Issue** | **Comments** |
| Number of SMS reminders sent as planned | • 1913/2395 (80%) of daily SMS<br>• 631/872 (72%) of weekly SMS |
| SMS reminders not delivered due to technical challenges (e.g., poor network coverage) | • 132/2395 (13%) of daily SMS<br>• 183/872 (21%) of weekly SMS |
| Number of SMS sent later due to technical issues such as poor network | • 161/2395 (7%) daily SMS<br>• 58/872 (7%) weekly |
| Number of triggered SMS sent | • 113/1356 (8%)<br>• 296/1397 (21%) |
| **Acceptability** | |
| **Preference of SMS type/frequency**[*]<br>Daily reminder versus triggered or weekly (N = 19)<br>Weekly reminders versus triggered or daily (N = 20) | 17 (89%) from the daily arm preferred daily SMS reminders to triggered SMS reminders<br>None of the participants who received daily SMS wished they received weekly reminders.<br>16 (80%) from the weekly arm preferred triggered SMS reminders to weekly<br>17 (85%) who received weekly reminders wished they were receiving daily reminders |
| Default reminder<br>Personalized reminders | 10 (26%) preferred default SMS text (i.e. "*This is your reminder*")<br>29 (74%) preferred personalized SMS text—reminders formulated by TB patients themselves (e.g. "*How are you today*", "*Take care*", "*Hello, have you had lunch*", "*Football*", "*How are you moving on*") |
| **Feasibility and acceptability of notifications to social supporters N = 39 (i.e. 19 in the daily SMS arm and 20 in the weekly SMS arm** | |
| Number of SMS notifications sent as planned | • 1577/2395 (66%) for daily SMS<br>• 740/872 (85%) for weekly SMS |
| SMS notifications not delivered due to technical challenges (e.g., poor network coverage) | • 639/2396 (27%) of daily notifications<br>• 45/872 (5%) of weekly notifications |
| Number of SMS notifications sent later due to technical issues such as poor network | • 180/2395 (8%) of daily notifications<br>• 87/872 (10%) of weekly notifications |

*Preference for SMS type/frequency was collected via patient questionnaires; all other data were collected from Wisepill device.

**Table 3. Feasibility and acceptability of real-time adherence monitor.**

| (N = 58, i.e. 19 in the daily SMS arm, 20 in the weekly SMS arm, 19 in the control arm) | |
|---|---|
| **Feasibility issue** | **Comments** |
| Data loss* | 342/9224 (3.7%) of data were lost because of technical issues with the adherence monitors. |
| Device malfunction* | 2 (3%) out of 60 devices malfunctioned due to technical failures and were replaced. |
| Device battery changes* Taking pills from another source *Reason(s) (n = 10)* •Was traveling •Had not refilled the device *Openings without pill removal (n = 8)* •During refilling the monitor •Opened accidently | Study staff replaced 4 batteries; poor mobile network resulted in repeated attempts to transmit the data which depleted the batteries before the anticipated battery lives. 10 (18%) of TB patients reported ever taking pills from another source during the study period. 7 (70%) 3 (30%) 4 (50%) 4 (50%) |
| *Acceptability* | |
| Perceived usefulness Perceived ease of use | 58 (100%) of TB patients reported that the real-time monitor: a) was useful in their TB medication adherence; b) helped them take their medication in time/as prescribed; d) positively affected the way they felt about taking medicine; e) and made it easier for TB patients to take their medication. 58 (100%) of TB patients found the real-time monitor easy to open and close, and remember to get pills. 45 (78%) of TB patients found the real-time monitor easy to travel with. |
| Disclosure from the real-time monitor | 3 (5%) of TB patients reported that the adherence monitor resulted in unwanted disclosure of their TB status |

*Data about data loss, device malfunction, and battery changes were collected from Wisepill device. All other data were collected via patient questionnaires.

supporters from both the daily (n = 14; 74%) and weekly (n = 15; 75%) SMS arms respectively preferred triggered SMS notifications to daily or weekly SMS notifications. Social supporters (n = 24; 62%) reported improved relationships with their TB patients (e.g., frequent communication with each other) as a result of participating in this study. Many social supporters (n = 25; 64%) reported instances of failing to provide the assistance required by the TB patients, mainly due to lack of money (n = 15; 60%).

*Real-time adherence monitor.* As indicated in Table 3, all TB patients (n = 58; 100%) reported that using the real-time monitor: a) was useful in their TB medication adherence including being more useful than the pill sackets often used for TB medication; b) helped them take their medication in time/as prescribed; d) positively affected the way they felt about taking medicine; e) and made it easier for TB patients to take their medication. All participants also found the real-time monitor easy to use (e.g., opening and closing). Ten (18%) of TB patients reported ever taking pills from another source during the study period during travel or when they had not refilled the monitor, while 8 (14%) of TB patients reported opening the monitor without taking medication during refilling the monitor as well as accidental openings.

**Acceptability of the technologies.** As shown in Table 4 below, results from exit interviews indicated that receiving an SMS reminder and looking at the monitor reminded TB patients take their medications on time. Additionally, TB patients reported that being sent an SMS reminder and being monitored implied care and motivated medication taking. Also, TB patients reported receiving social support (e.g., reminders, counseling) from social supporters in response the SMS notifications received. Challenges reported by participants included possibility of TB status disclosure as a result of using the technologies, unreliable networks, and lack of money.

**Table 4. Qualitative results demonstrating technology acceptability.**

| Theme | Example quotation |
|---|---|
| | *Perceived usefulness* |
| SMS reminders and real-time adherence monitor **reminded participants to take their medications** | *Whenever I would receive your SMS, immediately I would prepare and take my medication. ~Female, 33 years*, TB patient, *Daily SMS arm*<br>*R: I put the monitor near my bed, and when I look at it in the morning, I remember that one of the things I need to do in the morning is to take my drugs. ~ female, 33years*, TB patient, *Daily Arm A* |
| Receiving SMS reminders and being monitored **implied care** | *I would feel good after receiving an SMS because I know that you are caring about my life and want me to get well. So, I tried to take my drugs well so that you do not think that you are minding about someone who does not mind about his own life. ~ Male, 41years*, TB patient, *Weekly SMS Arm.* |
| Monitoring **encourages medication adherence** | *Because I knew you were monitoring the way I took my medication through the device, I made sure I was taking my medication because I did not want to disappoint people who cared about me. ~ Male 25 years*, TB patient, *Daily SMS Arm.* |
| SMS notifications to social supporters **enabled provision of social support** | *Whenever he (social supporter) would receive your message, he would call me to check how I was feeling and remind me take medication, and I would take my medication immediately. ~ Male, 41 years, Weekly SMS Arm.*<br>*Whenever he sees your message, he tells me to open the device and swallow my drugs before the computer detects that I have not swallowed the drugs. Whenever he starts telling me this, he won't stop until I have taken the drugs. ~ female, 56 years,* TB patient, *Daily SMS Arm* |
| | *Challenges* |
| Possibility of unintended TB status **disclosure** | *One time I went with it at my mother's house, and when she saw the monitor, she asked what it was and I had to explain, and in the process she ended up knowing that I had TB. I think if I had not carried the device with me, she would not have known. ~Female, 33 years*, TB patient, *Daily SMS Arm.* |
| **Inability to address structural barriers to non-adherence** | *When I got sick, I became too weak to continue working. So, sometimes when I have no food, and sometimes I have no money for transport to the clinic, thus, I still miss taking the medication even if I receive the SMS reminder ~ Female, 38 years,* TB patient, *Weekly SMS Arm B.* |

## Medication adherence

**Primary analysis.** The median adherence over the 6-month study period was 96.1%, (IQR 84.8–98.0), 92.5% (IQR 80.6–96.3), and 92.2% (IQR 56.3–97.8) respectively in the daily SMS, weekly SMS, and control arms and was similar, within study arm, over the two study phases. In the univariable analysis, average adherence was 8.4% (-5.8, 22.5; p = 0.24) higher in the daily SMS arm than the control arm while that in the weekly SMS arm was 4.7 (-10.3, 19.8; p = 0.53) higher than that in control arm. In the multivariable analysis controlling for gender, food security, HIV status social support and TB disclosure, participants in the daily SMS arm had 9.9% (95% CI -4.8, 24.5; p = 0.18) higher adherence than participants in the control arm, while those in the weekly SMS arm had 6.3% (95% CI -8.7, 21.3; p = 0.40) higher adherence than those in the control arm. The results were similar when considering the secondary data-set. In both the univariate and multivariate analyses, the higher adherence observed was not statistically significant as indicated by the P values.

## Discussion

Findings from this study show that real-time electronic monitoring and the use of SMS as reminders for TB patients and notifications for provision of social support were largely feasible and acceptable for TB medication adherence in southwestern Uganda. The pilot RCT revealed higher median adherence in the daily SMS and weekly SMS arms compared to the control, although the study was not powered for statistical significance when controlling for gender, food security, social support, stigma, and HIV status. These findings suggest preliminary evidence for clearly meaningful effect that should be considered for further evaluation in a fully powered larger trial.

The use of real-time adherence monitoring linked to SMS reminders to TB patients was feasible for supporting TB medication adherence in southwestern Uganda. All TB patients perceived the technologies to be easy to use. Most SMS reminders were sent as planned, demonstrating feasibility of this technology. The real-time medication monitors generally worked well; most of the adherence data was transferred as expected. Although no other studies have reported the feasibility of real-time adherence monitoring linked to SMS reminders for TB medication (studies such as the ASCENT trial, ClinicalTrials.gov number, NCT02574455 will be forthcoming), these technologies have been similarly reported to be feasible for monitoring ART adherence [17]. Despite the general feasibility of the technologies in the current study, poor mobile networks, not using the technology appropriately, and lack of money for transport to the clinic could limit the impact of the intervention. Notably, cellphone network is rapidly expanding in Uganda [36], and there is widespread cellphone adoption including among TB patients [37], suggesting these technical challenges may improve with time. "With 2G networks reaching nearly the entire population in Uganda, and with mobile broadband networks (3G/4G) covering more than 80% of the country [38], poor network is not a significant challenge. Although reported by few participants, opening the monitor without taking medication or taking medication from other sources could limit the accuracy of adherence monitoring. Some of this bias can be accounted for in analysis (e.g. excluding more openings per day than would be expected).

The use of social support notifications to social supporters was feasible—most notifications were sent as planned, and all social supporters reported assisting the participants (e.g. transport to the clinic, money for buying food and drinks, counseling) at least once after receiving SMS notifications. Social support (such as food supplements, and economic support) delivered through face-to-face approaches improved TB treatment completion in various settings including low resource settings such as Nepal, Burkina Faso, and Haiti [32]. However, the impact of SMS notifications could be constrained by the reported lack of financial resources to support TB patients. Complementing the SMS notification intervention with interventions to boost the economic status [39] of social supporters (e.g. through income generating activities such as farming) may improve their capacity to provide the support needed by the patients.

The use of real-time adherence monitoring linked to SMS reminders to TB patients and notifications to social supporters was acceptable for supporting TB medication adherence in this setting. All TB patients perceived the technologies to be useful in supporting TB medication adherence. Receiving SMS reminders and being monitored reminded TB patients to take their medication on time, thus, addressing the challenge of forgetfulness. This function is especially important for TB patients with busy schedules or those inexperienced with regular medication. Additionally, receiving SMS and being monitored with a real-time monitor were perceived as being "cared for". This perception can encourage TB patients to take medication in order to cooperate with those who gave them the intervention while at the same time taking care of their own lives. Notably, the use of these technologies to support ART

medication adherence was reported to be acceptable among people living with HIV in the same setting [17].

Overall, our findings indicate that real-time adherence monitoring linked to SMS reminders to TB patients and notifications to social supporters is feasible and acceptable for supporting TB drug administration. This is specifically critical in settings like Uganda where the implementation of DOT has been abandoned mainly due to the financial burden it places on the TB patients (in the form of transport to the clinic for drug administration), as well as time demands on healthcare workers (who have to physically watch TB patients take their medication amidst other competing demands). This technology can keep TB patients connected with healthcare supporters, which can potentially minimize the challenge of social isolation that is often associated with TB.

Our study found that SMS reminders may improve TB medication adherence. Compared to weekly or triggered SMS, TB patients found daily reminders matching well with their daily pill taking and perceived them as daily encouragements to take their medications. The prior study in the same setting showed that daily SMS reminders integrated with real-time monitoring can improve ART adherence [18]. High quality studies utilizing SMS reminders (not linked to real-time adherence monitoring) to support TB medication report mixed results. A recent systematic review of 14 randomized controlled trials and quasi-experimental studies done prior to 2019 reported essentially no difference in TB treatment success (i.e. pooled estimate of 87% with SMS vs 85% in the control group) [40]. In another recent systematic review of 16 randomized controlled trials that utilized digital health technologies to support TB medication adherence, only 2 of the 8 studies that utilized interactive SMS reminders and educational SMS texts reported positive benefits of using this technology in reducing medication adherence defaults and increasing cure rate in China [41] and treatment adherence and cure rate in Argentina [42]. Compared to our SMS reminder system that was one-way, automatic and linked to a real-time technology, the study in Argentina was more interactive and had a common feature of providing a sense of regular support. The remainder of the SMS reminder studies (not linked to a real-time technology) found no effect of SMS reminders on TB medication adherence compared to the standard of care in United States, Spain, Hong Kong, and South Africa [43–47]. British Columbia Canada [44], Sudan [45], Pakistan [48], and Cameroon [49]. In the same review, the use of real-time adherence monitors integrated with medication reminders resulted in fewer missed doses compared to real-time adherence monitors without reminders in China [19], while real-time adherence monitors without reminders reduced TB treatment defaults in Haiti [50]. The variable results among studies of SMS reminders highlight the importance of understanding how the technology is used and perceived. The causes of the differences in findings are unclear but might relate to context, culture, degree of technology exposure, TB patients involvement in technology development, fidelity of technology use, varying measures of adherence, access to social support, and other factors such as depression, and lack of transport to the clinic.

The acceptability of the technologies can be limited by the possibility of unintended TB status disclosure (e.g., when someone sees the SMS or monitor) which might result into stigma and discrimination. Using messages that are not easily linked with TB as preferred in this study can reduce the possibility of SMS-enabled unintended status disclosure. It should however be noted that technology-enabled disease status disclosure is not always unwanted; for instance, participants living with HIV in the same setting had previously reported using the monitor to intentionally disclose their HIV status mainly to enable them get social support [51]. For TB patients who would want to keep their status private, the monitor-led unwanted status disclosure could be minimized by redesigning it to better meet the needs of TB patients (e.g., making the monitor more portable and discreet). Additionally, interventions that reduce stigma can be incorporated.

In the current study, many social supporters reported improved relationship with TB patients as a result of using the SMS notifications. Social supporters might have felt more obliged to be there for TB patients after having been identified (by TB patients themselves) as their helpers (who should receive notifications), in order not to disappoint them. TB patients might have felt the need to meet the expectations (i.e., adherence) of social supporters to not to undermine the support provided and to keep getting the support [52]. This finding is in contrast to that reported in a related intervention for supporting ART adherence in the same setting, in which some conflicts arose as a result of mismatch of expectations between TB patients and their social supporters [25] and lack of clear benefits of social support [18]. Formative results from the current study highlighted the possibility of misunderstandings between social supporters and TB patients as a result of the frequent receipt of SMS notifications which may be perceived as lack of commitment to taking medication [53]. The possibility of such misunderstandings was managed through orientation.

This study has some limitations. From qualitative work, we found out the real-time monitor alone can motivate TB patients to take medication. Therefore, using the real-time monitor in the control arm could have had its own intervention effect, which might have affected the detection of intervention effects in other study arms. Although the use of the real-time adherence monitor may be more accurate than traditional approaches to adherence monitoring (such as self-report and clinic-based pill counts), its non-use (e.g., taking drugs from other sources during travel) and/or inappropriate use (e.g., opening it without taking medication) might have affected the accuracy of adherence estimations of a few TB patients across study arms. Also, for this pilot study, rather than intention to treat analysis, we used per protocol analysis that involved excluding TB patients who voluntarily withdrew from the study or declined to use the adherence monitor. Being a pilot study, the small sample size involved limits results generalizability. Lastly, since participants self-reported their perceptions about the intervention, qualitative results (and some self-reported quantitative aspects) could be vulnerable to social desirability bias especially since participants knew that they were reporting on the intervention given to them by the same researchers collecting data from them.

To the best of our knowledge, this is the first study to investigate the use of real-time adherence monitoring linked to SMS reminders to TB patients and notifications to social supporters among TB patients. Real-time monitoring creates the potential for intervening prior to the development of treatment failure and/or drug resistance. Our multiple approaches to the use of SMS (i.e. daily, weekly, triggered) made it possible to identify TB patients' preferences and highlight the usefulness of these approaches.

In summary, we found that real-time adherence monitoring linked to daily SMS reminders and notifications followed by triggered SMS reminders and notification is generally feasible. SMS were sent and received as planned, and data from the monitor was transferred as expected, most of the time. The technology was acceptable and feasible in a resource limited setting. The intervention appeared to improve TB adherence, though the difference in study arms was not significant; importantly, this study was not powered for differences in medication adherence. Larger studies are needed to determine the larger-scale feasibility, acceptability, and impact on TB treatment adherence and clinical outcomes. Integrating economic support to the intervention could empower social supporters to meet the financial needs of TB patients, which they would otherwise not be able to meet; such work is ongoing (NCT05656287). Future work should also analyze the cost of the technology to inform potential scalability.

## Supporting information

**S1 Checklist. Reporting checklist for randomised trial.**
(DOCX)

**S2 Checklist.**
(DOCX)

**S1 Appendix. Univariable and multivariate analysis of adherence across arms.**
(DOCX)

**S1 File. Trial protocol.**
(DOC)

**S1 Text. Questionnaire packet—study title: Real time tuberculosis medication adherence intervention in rural southwestern Uganda.**
(DOC)

## Acknowledgments

The authors would like to acknowledge the contributions of Wisepill Technologies and study participants.

## Author Contributions

**Conceptualization:** Angella Musiimenta, David Bangsberg, J. Lucian Davis, Jessica E. Haberer.

**Data curation:** Nicholas Musinguzi.

**Formal analysis:** Nicholas Musinguzi.

**Funding acquisition:** Angella Musiimenta.

**Investigation:** Angella Musiimenta, Wilson Tumuhimbise, Esther C. Atukunda, Aaron T. Mugaba, Conrad Muzoora, David Bangsberg, J. Lucian Davis, Jessica E. Haberer.

**Methodology:** Angella Musiimenta, Nicholas Musinguzi, J. Lucian Davis, Jessica E. Haberer.

**Project administration:** Angella Musiimenta, Wilson Tumuhimbise, Aaron T. Mugaba.

**Resources:** Angella Musiimenta, Wilson Tumuhimbise.

**Software:** Wilson Tumuhimbise, Nicholas Musinguzi.

**Supervision:** Angella Musiimenta.

**Validation:** Angella Musiimenta, Jessica E. Haberer.

**Visualization:** Wilson Tumuhimbise, Aaron T. Mugaba, Nicholas Musinguzi.

**Writing – review & editing:** Angella Musiimenta, Wilson Tumuhimbise, Esther C. Atukunda, Nicholas Musinguzi, David Bangsberg, J. Lucian Davis, Jessica E. Haberer.

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
