## [Decision Letter · Decision Letter 0]

25 Apr 2023

PGPH-D-23-00372

The Feasibility, Acceptability, and Preliminary Impact of Real-Time Monitors and SMS on Tuberculosis Medication Adherence: Findings from a Pilot Randomized Controlled Trial

Dear Dr. Musiimenta,

Thank you for submitting your manuscript to PLOS Global Public Health. After careful consideration, we feel that it has merit but does not fully meet PLOS Global Public Health’s publication criteria as it currently stands. Therefore, we invite you to submit a revised version of the manuscript that addresses the points raised during the review process.

We look forward to receiving your revised manuscript.

Kind regards,

Dr. Lucy Chimoyi, PhD

Academic Editor

Journal Requirements:

1. Please provide separate figure files in .tif or .eps format only and remove any figures embedded in your manuscript file. Please also ensure that all files are under our size limit of 10MB.

4. In the online submission form, you indicated that "The datasets analysed during the current study are available from the corresponding author on reasonable request". All PLOS journals now require all data underlying the findings described in their manuscript to be freely available to other researchers, either 1. In a public repository, 2. Within the manuscript itself, or 3. Uploaded as supplementary information.

Additional Editor Comments (if provided):

1.The qualitative data collection tools were developed using the UTAUT. I am assuming that the analysis and presentation of  results followed this theory. However, this is not clear from Table 4. Include the theory used as part of the caption. 

2. It is advisable for authors to include in their title that this manuscript followed a mixed-methods approach.

3. A sub-section on outcome measures in the methods section should be included and all outcome measures defined (feasibility, acceptability and adherence).

4.From this work, do you think that poor network coverage is a problem or a limitation? This has been mentioned in the results, but authors have not discussed its implication. Is it something that implementers need to be concerned about?

5. Thank you for including the CONSORT checklist, also include the COREQ checklist

6. Was saturation determined during analysis of the qualitative data?

Reviewers' comments:

Reviewer's Responses to Questions

**Comments to the Author**

1. Does this manuscript meet PLOS Global Public Health’s publication criteria? Is the manuscript technically sound, and do the data support the conclusions? The manuscript must describe methodologically and ethically rigorous research with conclusions that are appropriately drawn based on the data presented.

Reviewer #1: Yes

Reviewer #2: Yes

Reviewer #3: Yes

2. Has the statistical analysis been performed appropriately and rigorously?

Reviewer #1: Yes

Reviewer #2: Yes

Reviewer #3: Yes

3. Have the authors made all data underlying the findings in their manuscript fully available (please refer to the Data Availability Statement at the start of the manuscript PDF file)?

Reviewer #1: No

Reviewer #2: No

Reviewer #3: No

4. Is the manuscript presented in an intelligible fashion and written in standard English?

Reviewer #1: Yes

Reviewer #2: Yes

Reviewer #3: Yes

5. Review Comments to the Author

Reviewer #1: This article reports the feasibility and acceptability of a piloted TB adherence support strategy that combines a medication reminder box and SMS reminders to both patients and supporters. The study is a valuable addition to the TB digital adherence technology literature, but there are several aspects where I believe that there is incomplete reporting or a lack of transparency/clarity in the methods and results.

1) The second paragraph of the introduction refers to DOTS, which I find confusing because of the way this acronym has been used by WHO. The authors are discussing the practice of directly observed therapy (commonly abbreviated DOT), and I agree with everything they say about its shortcomings. However, the authors use the abbreviation DOTS (with the terminal S), which is generally used to refer to the old WHO DOTS Strategy. This strategy was included much more than DOT. It had five components - government commitment, a reliance on smear microscopy, directly observed short-course treatment, uninterrupted drug supply, and standardized reporting; even the third component (which corresponds to the phrase “directly observed therapy – short course” mentioned by the authors) encompassed not only the practice of DOT but the specific regimens that were recommended at the time. I think the authors should make it clear that they are talking about the practice of DOT, not the DOTS Strategy. This confusion could be resolved by simply defining the phrase “directly observed therapy (DOT)” in the first sentence of the paragraph, and using the abbreviation DOT rather than DOTS throughout.

2) In the statistical analysis section, I am a little confused by the use of the term “censoring.” My understanding of the term censoring in statistics is that it is used (usually for survival analysis) when partial information is known about the outcome, specifically that the event of interest has not yet occurred by a certain point, but that it will happen in the future. If I am understanding the analysis correctly, the authors did not use a survival analysis – I believe they used a linear regression with adherence as a continuous outcome. There are various periods of time that are excluded from the adherence calculation (e.g. time after death), but I do not think that this is censoring because the information about the outcome is not in doubt (participants could not have been adherent after death), and the regression model is not treating it as such. Maybe it would be clearer to just say that these days were excluded from the adherence calculation (i.e. excluded from both numerator and denominator)?

3) A definition of unnecessary SMS (reported in Table 2) and how it was determined should be included in the methods.

4) I suggest that the authors include the questionnaire as a supplementary file. It is unclear how some of the information in Tables 2 and 3 and the text was elicited (specifically the sections “SMS reminders and notifications”, “SMS notifications to social supporters”, and “real-time adherence monitor”). For example, how were patients asked about their preference for daily/weekly versus triggered SMS reminders given that they presumably received both? Do the responses mean that they would have wanted *only* the triggered reminders? How was ease of use assessed – a binary answer of “easy” or “difficult” or a Likert scale? Was fear of disclosure specifically asked about, or did 5% report this unintended consequence in the open-ended part of the questionnaire? What other forms of support could have been reported by treatment supporters.

5) Tables 2 and 3 mix indicators that were collected via the device, as well as things that I assume the authors could only know by asking the patient, and things that may have been recorded through the study teams’ field notes. This makes it a little confusing to evaluate, and it makes me wonder why these particular indicators are being reported in the tables. One way to improve transparency would be to add a column or footnotes to the table indicating the source of each data element. Another would be to include the questionnaire as supplementary information as noted above.

6) How many subjects participated in interviews? Did the interviewees know that the interviewers were representatives of the study that provided the monitors (raising the possibility of social acceptability bias in responses)? (I am not going to go down the COREQ checklist and raise every point that is not included in the methods, but I think that these two points are particularly important for judging validity).

7) In the qualitative interviews, were there minority opinions of participants who contradicted the general themes included under “perceived usefulness”? For instance, was there anyone who said that the medication reminder was annoying and that they would have remembered to take their meds anyway since they are used to taking daily ART? The qualitative results reported are rather scanty, and all but one of the themes reported about the technology itself are positive. This raises the concern that negative or ambivalent opinions are not being reported to make a case for acceptability.

8) The second paragraph of the discussion is confusing in that the first sentence claims, “the use of social support notifications to social supporters was feasible,” while the third sentence says “the feasibility of SMS notifications to social supporters in our study was limited.” The following sentences then talk about the need to improve feasibility. I think that the authors are either mixing the discussion of feasibility of two different things, or, starting with the third sentence, they are actually making a different point that is not about feasibility. The first sentence is clearly talking about the feasibility of delivering SMS notifications to social supporters. The remaining sentences are discussing whether these notifications are likely to have an impact given the limited ability of social supporters to overcome socioeconomic barriers. To me, this is a different point that is not about feasibility. However, the authors may be conceptualizing the point as a broader discussion about the feasibility of a hypothetical social support intervention that could utilize the SMS messages – not the actual intervention they piloted. I am fine with either conceptualization, but it has to be clear *what* is feasible or not.

Minor:

Line 342 reports that “participants who had disclosed their status” had lower adherence. What statis was being disclosed and to whom?

Line 443 – I think the authors mean discreet, not discrete

Reviewer #2: Thank you authors for this informative manuscript. Kindly find herewith some suggestions and questions that would help improve this work.

Line 167 - 168 ? -- What criteria was used by the Wisepill to determine acceptability and feasibility?

Line 263 "Overall, a total of 39 social supporters completed the study"

line 266 "Of the 42 social supporters"

Line 295 "All social supporters (n = 39; 100%)

Please clarify the correct number of social supporters analyzed.

line 259 - "..totaling to 63 PLTB"

Line 269 Table 1 - baseline demographics characteristics shows (N=63)

Line 307 -- "All PLTB completing the study (n=58; 100%)"

Please clarify the correct number of PLTB analyzed.

Line 315; Table 3, Reason(s) (n = 10)

However, the corresponding total number of the participants in the three reasons is greater than 10 i.e. (7+3+8 = 18)

Deeper analysis based on stratifying the TB treatment phases; intensive phase (2 months) and continuation phase (4-6 months) based on the frequency of reminders could further inform the non-adherence

During the qualitative data analysis, did any patient raise the issue of SMS reminders fatigue due to the daily frequency of incoming messages as reported in other studies? JAMA Intern Med. 2016;176(3):340-349. doi:10.1001/jamainternmed.2015.7667

Overall grammatical corrections to improve readability

Reviewer #3: Congratulations to this team for a fantastic study and a comprehensive and interesting evaluation of real-time adherence monitoring. The combination of qualitative data provides a nice overview of the benefits and potential drawbacks of this approach. Below I list a number of issues, nearly all minor, that should be clarified.

Author affiliations – no one is assigned 5 or 6.

Abstract: please specify primary outcomes. Was adherence a primary outcome?

Abstract: Please add confidence intervals around adherence percentages – remove reference to “may have improved adherence” as there is little evidence cited in the abstract for this. Alternatively cite the adjusted adherence findings with confidence intervals and could leave this in. See my note at the end regarding statistical significance.

Throughout the paper there seems to be a conflation of DOTS (the broader 5 tenet strategy) and DOT (directly observed treatment), which is an element of DOTS. To my read, DOTS is used in places where DOT, but not DOTS, would be appropriate.

Line 93 – minor typo “bn”

Page 5 refers to formative findings from the current study and then a formative study. Which is correct? Or are these references to two different studies?

Line 135 p6: Change “cater for” to “account for”

P6, line 138 – what does it mean to recruit them at random? Was this a convenience sample? If not, explain the systematic for recruitment

Line 143 – Explain how access to cellular network was assessed.

Not clear how purposive sampling fits in to the design. To whom does this apply? People with TB? Supporters? Both?

Could a patient participate if they did not have a supporter?

How often and at what frequency were SMS reminders sent to supporters? Was it the same as for the PLTB?

Primary outcomes are defined as adherence but as the authors note, there was insufficient power for these outcomes. Were these really the primary outcomes or were the primary outcomes feasibility and acceptability?

Line 193 – what do the authors mean by non-responsiveness?

When was the data censored in patients with LTFU? At the end of the two months without treatment or the beginning of those first two months?

Line 223 – perhaps “secondary data analysis” instead of “dataset”?

Why were HIV and social support, versus other variables, considered a priori confounders? Especially in the context of a trial?

Line 263 – 39 of 45 completed the study but the text only explains that 3 of the 45 were excluded. What happened to the other 3?

Nearly 25% of messages were not received – this demonstrates feasibility and room for improvement; worth highlighting this in the discussion

Table 1 is very nice.

Table 2 is also nice, but it would be useful to have the percentage of daily, weekly and triggered messages received rather than the percentage of messages received that were daily, weekly, triggered. Same with messages not sent.

In tables – use Severely Food Insecure or Severe Food Insecurity

Possible to differentiate between messages not sent and those that were delayed – it seems like this is an important distinction.

Could provide more info on unnecessary messages

Lines 297-299 is difficult to understand – could be reworded for clarity

Lines 302 – 305 could be further explained.

Table 3 - Opening the monitor without taking pills is in the wrong place I think

Qualitative table: Lack of money – is this the right title for this theme? It doesn’t relate to the technology. A more suitable title might be “inability to address structural causes of non-adherence” or something along those lines. Adherence monitoring can overcome forgetfulness and perhaps mitigate others, but is unlikely to triumph over extreme poverty.

Lines 333/3 – please add labels to the numbers in parentheses. Are these confidence intervals? Ranges?

There is an adjusted analysis in lines 335/6. The results of an equivalent, unadjusted analysis should precede the adjusted analysis.

Disclosure is adjusted for but not included in the table 1.

Suggest removing the findings related to gender, and food insecurity because the analyses are underpowered, these findings are outside the objective of this analysis and you did not look at specific confounders for these variables so these estimates cannot be interpreted in the same way as your main exposure variables (study arm).

Throughout the paper, suggest de-emphasizing lack of statistical significance. You weren’t adequately powered for statistical significance. Instead, would highlight preliminary evidence for a potential clinically meaningful effect that should be evaluated in a larger trial

Please add reference to ASCENT trial or clinicaltrial.gov number

369 – not sure the adherence monitoring via hair/urine really fits in here. I don’t think you are suggesting that these should be used programmatically. What are you are suggesting? Consider removing.

381 (devise, not device); also “farming” versus “faming”

Worth one final close proofread for typographical edits

6. PLOS authors have the option to publish the peer review history of their article (what does this mean?). If published, this will include your full peer review and any attached files.

**Do you want your identity to be public for this peer review?** For information about this choice, including consent withdrawal, please see our Privacy Policy.

Reviewer #1: No

Reviewer #2: No

Reviewer #3: No

---

## [Decision Letter · Decision Letter 1]

20 Jun 2023

PGPH-D-23-00372R1

The Feasibility, Acceptability, and Preliminary Impact of Real-Time Monitors and SMS on Tuberculosis Medication Adherence: Findings from a Mixed Methods Pilot Randomized Controlled Trial

Dear Dr. Musiimenta

Thank you for submitting your manuscript to PLOS Global Public Health. After careful consideration, we feel that it has merit but does not fully meet PLOS Global Public Health’s publication criteria as it currently stands. Therefore, we invite you to submit a revised version of the manuscript that addresses the points raised during the review process.

There has been substantial improvement of the paper and we appreciate this work. The paper is much closer to publication. Please see the comments and edits outlined by Reviewer 1. These comments would best guide the changes that are needed.

We look forward to receiving your revised manuscript.

Kind regards,

Megan Coffee, MD, PhD

Academic Editor

Journal Requirements:

Additional Editor Comments (if provided):

Reviewers' comments:

Reviewer's Responses to Questions

**Comments to the Author**

1. If the authors have adequately addressed your comments raised in a previous round of review and you feel that this manuscript is now acceptable for publication, you may indicate that here to bypass the “Comments to the Author” section, enter your conflict of interest statement in the “Confidential to Editor” section, and submit your "Accept" recommendation.

Reviewer #1: (No Response)

Reviewer #2: (No Response)

2. Does this manuscript meet PLOS Global Public Health’s publication criteria? Is the manuscript technically sound, and do the data support the conclusions? The manuscript must describe methodologically and ethically rigorous research with conclusions that are appropriately drawn based on the data presented.

Reviewer #1: Partly

Reviewer #2: Yes

3. Has the statistical analysis been performed appropriately and rigorously?

Reviewer #1: N/A

Reviewer #2: Yes

4. Have the authors made all data underlying the findings in their manuscript fully available (please refer to the Data Availability Statement at the start of the manuscript PDF file)?

Reviewer #1: No

Reviewer #2: Yes

5. Is the manuscript presented in an intelligible fashion and written in standard English?

Reviewer #1: Yes

Reviewer #2: Yes

6. Review Comments to the Author

Reviewer #1: Thank you to the authors for their revision. However, most of my concerns from the original review have not been addressed. In addition, now that I am able to see the questionnaires, I have some additional concerns (which I alluded to in the original review but was not able to assess without seeing the data collection forms).

1) My question about participants knowing investigators has not been addressed. Self-report does not alone lead to social desirability bias; knowing that the investigators are the ones who gave them the monitor does. (For example, if the Ministry of Health gave out the monitors and everyone hated them, and the investigators were perceived as being from an organization not aligned with the Ministry of Health, then people might feel perfectly comfortable complaining about them, so it would still be self-report but there would be less risk of social desirability bias).

2) The COREQ checklist has not been applied properly, and some of the information is still not in the paper. First, the checklist is supposed to reflect the qualitative component of the study, and therefore the information should only pertain to the interview procedures and participants, not the overall study population. Second, the information is supposed to be in the paper, not in the checklist (the checklist just says where to find it in the paper). The purpose of the checklist is to ensure that there is sufficient information in the paper to allow the reader to judge robustness and bias of the qualitative results.

3) My concerns over the lack of clarity in the tables about where information comes from have to some extent been clarified by attaching the questionnaires. However, seeing the questionnaires has not satisfied my concern about the authors having picked certain indicators to report and not others, and it makes me worry that they have cherry-picked the numbers that make the technology look acceptable. For example:

- Why does Table 2 report the number (%) of people in the weekly reminder group who would have preferred daily reminders but not the number (%) of people in the daily reminder group who would have preferred weekly reminders?

- Table 3 has “Perceived ease of use” reported as a n (%), but as far as I can see in the questionnaire, this question was asked as a Likert item, so it is not obvious what the reported value corresponds to.

- Also, the questionnaire actually asked four different questions about ease of use (opening, remember how to get pills, charging, traveling), so it is unclear why the authors only reported the first.

- The text at line 322 says “As indicated in Table 3, all PLTB completing the study (n=58; 100%) perceived the real-time monitor to be useful in reminding them to take their medications and implying care by healthcare workers.” However, this information is not in Table 3, which only reports ease of use, not usefulness. And I do not know what data support the statement “implying care by healthcare workers.”

4) The methods mention that the questionnaires and interview guides were developed based on the Unified Theory of Acceptance and Use of Technology model, which is one of the scientific strengths of this study. Looking at the questionnaire, the “Technology Adoption” section clearly reflects the UTAUT model and I think is the strongest aspect of the questionnaire since it is well grounded in a behavioral theory that can help interrogate the drivers of acceptability and adoption. However, the results from this section of the questionnaire are not really reported, except for isolated questions that have been picked for unclear reasons and collapsed into binary responses. I do not know why the authors chose to ignore this important section of their questionnaire – I realize it is possible that they are saving it for another paper. However, if that is the case, then I would not claim the UTAUT conceptual framework in the methods because the use of the framework is not at all evident in the results.

5) My concern about specifying data sources for the tables has been inadequately addressed. The Table 2 footnote should specify where all the data come from; I would suggest expanding the footnote to say, “Preference for SMS type/frequency was collected via patient questionnaires; all other data were collected from Wisepill device” (if that is the case). Table 3 should also have a similar footnote indicating where data came from.

6) I still do not understand what a message “sent unnecessarily” means. I understand that there were delayed messages when network connectivity failed, but why does that make the message unnecessary?

7) I do not understand added phrase at line 330 (“manifested inform of frequent communications)

Reviewer #2: Thank you authors for this updated manuscript. However, some things are still not that are not clear. The following suggestions and questions might further help in improving this manuscript.

Introduction

The current paper reports on the feasibility, Acceptability, and Preliminary Impact of Real-Time Monitors and SMS on Tuberculosis Medication Adherence: Findings from a Mixed Methods Pilot Randomized Controlled Trial [both the title and introduction line 119] - However, the study definition and metrics of the feasibility and impact are not stated or explained in the writeup.

Line 35 - Feasibility - What was its definition? How was this it assessed?

Line 171 - The primary outcome was .. kindly clarify on this, a couple of outcomes are given.

Impact - How was this it assessed?

Line 177 - the outcome measure of the acceptability of the intervention is not clear.

Line 40: "Among 66 participants " -- Update this to reflect 63 PLTB as indicated on line 269, line 277 and table 1 (line 281)

7. PLOS authors have the option to publish the peer review history of their article (what does this mean?). If published, this will include your full peer review and any attached files.

**Do you want your identity to be public for this peer review?** For information about this choice, including consent withdrawal, please see our Privacy Policy.

Reviewer #1: No

Reviewer #2: No

---

## [Decision Letter · Decision Letter 2]

25 Sep 2023

PGPH-D-23-00372R2

The Feasibility, Acceptability, and Preliminary Impact of Real-Time Monitors and SMS on Tuberculosis Medication Adherence: Findings from a Mixed Methods Pilot Randomized Controlled Trial

Dear Dr. Muslimenta: 

Thank you for submitting your manuscript to PLOS Global Public Health. After careful consideration, we feel that the paper is almost ready for publication and has merit but does not fully meet PLOS Global Public Health’s publication criteria as it currently stands. Therefore, we invite you to submit a revised version of the manuscript that addresses the points raised during the review process.

The paper is close to ready for publication. I do not want to slow down publication but wanted to give you time to incorporate the helpful advice shared by Reviewers 4 and 5. If you are able to share the data with publication that would also be helpful.

Please submit your revised manuscript by October 6, if you are ready. You can take more time if needed, but I just wanted to stress that the paper is almost ready for publication. If you will need more time than this to complete your revisions, please reply to this message or contact the journal office at globalpubhealth@plos.org. Please include the following items when submitting your revised manuscript:

We look forward to receiving your revised manuscript.

Kind regards,

Megan Coffee, MD, PhD

Academic Editor

Journal Requirements:

Additional Editor Comments (if provided):

Reviewers' comments:

Reviewer's Responses to Questions

**Comments to the Author**

1. If the authors have adequately addressed your comments raised in a previous round of review and you feel that this manuscript is now acceptable for publication, you may indicate that here to bypass the “Comments to the Author” section, enter your conflict of interest statement in the “Confidential to Editor” section, and submit your "Accept" recommendation.

Reviewer #1: All comments have been addressed

Reviewer #2: All comments have been addressed

Reviewer #4: (No Response)

Reviewer #5: (No Response)

2. Does this manuscript meet PLOS Global Public Health’s publication criteria? Is the manuscript technically sound, and do the data support the conclusions? The manuscript must describe methodologically and ethically rigorous research with conclusions that are appropriately drawn based on the data presented.

Reviewer #1: Yes

Reviewer #2: Yes

Reviewer #4: Yes

Reviewer #5: Yes

3. Has the statistical analysis been performed appropriately and rigorously?

Reviewer #1: Yes

Reviewer #2: Yes

Reviewer #4: Yes

Reviewer #5: Yes

4. Have the authors made all data underlying the findings in their manuscript fully available (please refer to the Data Availability Statement at the start of the manuscript PDF file)?

Reviewer #1: No

Reviewer #2: Yes

Reviewer #4: Yes

Reviewer #5: Yes

5. Is the manuscript presented in an intelligible fashion and written in standard English?

Reviewer #1: Yes

Reviewer #2: Yes

Reviewer #4: Yes

Reviewer #5: Yes

6. Review Comments to the Author

Reviewer #1: All of my comments have been addressed

Reviewer #2: (No Response)

Reviewer #4: Overall the authors report on the results of a small pilot RCT to assess the feasibility and acceptability of weekly vs daily vs standard of care SMS reminder with real-time adherence monitoring plus social support using an implementation science informed approach complete with relevant framework/theory. While these are important considerations, the study is hampered by the small sample size and the concurrent strategy components being evaluated (timing, type of message, social support, etc). While the study shows some interesting trends it is a bit hampered by the small sample size.

1. In terms of implementation outcomes, in addition to acceptability and feasibility, was reach considered. One challenge with DAT is its potential with perpetuating inequity. Especially as the study excluded individuals who were unable to demonstrate mobile/cellphone use/ownership or signal or ability to maintain and charge, these may be associated with sociodemographic characteristics. While the authors share some content analysis related to underlying SDoH,

2. Did the study support/provide airtime for participants in receiving messages? If not, why not? Is this a sustainable model?

3. How was the threshold for feasilibility of SMS delivery determined. 66% fidelity to SMS daily while in theory is a “majority,” operationally does not seem reliable enough to consider it feasible.

4. In terms of the Data Availability Statement, I am unclear about whether requests through the origin IRB constitutes “absence of restriction” as denoted by the Journal’s policy.

5. Finally, some of the statements regarding the results of the univariate and multivariate analyses seem a bit out of proportion to the statistical tests reported. It would be important to note that “trends” of higher adherence are observed, but are not necessarily statistically significant based on the p values described. I encourage the authors to caveat their findings a bit more, especially given the small sample size.

6. In the discussion (paragraph 2), many suggestions are made for additional implementation strategy components that more directly address underlying social and structural determinants as barriers for TB affected individuals. However, the logic around increasing “feasibility” of the SMS to social supporters by providing these additional targeted components (financial resources, livelihood support, etc) is unclear. Instead of positioning these novel strategies as a way to increase the feasibility of the SMS, it seems more appropriate to state that these would increase the primary outcome- adherence, and to acknowledge that they may do that with or without the SMS intervention.

7. In conjunction with comment above, it also might be more appropriate to state that the SMS intervention is not designed to target social and structural determinants of health and so a future consideration would be linking this strategy to one that does in order to maximize not just feasibility but overall effect.

8. Paragraph starting line 415 requires references.

9. Discussion paragraph starting at line 424 could be improved by differentiating the reviews/prior literature on real-time adherence monitoring linked to SMS vs other SMS reminder systems.

Reviewer #5: The authors report on the results of a pilot randomized controlled trial among patients initiating treatment for drug-sensitive TB in a low-resource setting with high HIV co-infection in Uganda. It was a three arm ‘pilot’ trial, based on prior informative studies, whereby all participants received real-time electronic adherence monitors and were randomized (1:1:1) to daily or weekly SMS reminders to both patients and their identified social supporters for the first 3 months followed by only triggered SMS sent when the pill bottle was not opened for the remainder of the treatment course, vs a control arm that did not use SMS but dose monitor bottles only. They assessed acceptability, feasibility, and percent adherence levels by the real-time monitors as a primary outcome. As a 'pilot' RCT, it was not powered to detect a meaningful clinical outcome. Overall, this appears to be a well conducted study and is well presented. The inclusion of text messages to social supporters is interesting and appears to be an advancement over previous versions these authors have studied in terms of integrated adherence support linked to the monitoring. This use of the dose monitor plus texting approach in TB and including social supporters in receiving SMS messages is novel and should be of interest to people running TB programs and looking for alternative to in-person DOT. There are no major weaknesses I am concerned about, but a few minor suggestions.

Minor

Title/Abstract:

The title and/or abstract should include the study location/country for better context.

People/persons – suggest choose one. Also, PLTB often stands for Pleural TB in the existing literature, if not a standard acronym, perhaps avoid it or select a unique one.

One concern is that if SMS messages are used as ‘reminders’ for daily dosing, then 80% ‘sent’ rate would not be considered adequately feasible, as 20+ % lack of reminders may negatively influence adherence rates if they are in fact relied upon as reminders. If >80% of adherence is an expected minimum level for clinically effective treatment, this already starts at the bottom end? Also, if 96% of the texts were sent to social reports about adherence data, but this is more than the success rate to the patients, its hard to understand how the not sent messages to patients would be reported to social supporters? That said, the higher adherence levels in the daily SMS vs other arms is reassuring. Perhaps this can be discussed.

If all PLTB as stated found the intervention useful, then why and what additional features need to be included to overcome concerns of TB status disclosure and effects of poverty? I’m not sure this is a data derived statement.

Line 60 – Clarify “Only real-time monitoring creates the potential for intervening prior to the development of treatment failure and/or drug resistance” Clearly there are other ways to assess adherence and how the patients are doing prior to treatment failure or drug resistance.

Introduction:

Overall, it is well written and justified.

Line 82: Please provide a level and reference for the key statement “treatment adherences remain a challenge”, since this is core to the purpose of the study.

Study Design and Setting are fairly well described, but please include more details on standard of care adherence support, since this is critical context to the additional or interactive effects of the study interventions. For instance, what adherence training or support is given upfront. And if patients are seen in clinic/person every 2 weeks, what is the purpose of the weekly ‘reminder’ (not dosing which is daily, and not appointments, which are biweekly?).

Inclusion criteria – includes only people that can text (see below re estimating cellphone access and ability). If purposive sampling was used, would the assessed/screened represent the whole clinic population or was there screening bias? This is important in estimating accessibility for the intervention in the whole clinic context.

Line 153: please indicate which way purposive sampling was used to skew gender from clinic norms, and any other factors that may have influence recruitment bias. Consecutive would provide better whole populations estimates for factors related to inclusion criteria.

The Wisepill technology is fairly well described. Perhaps discussion of if separate SIM cards are required or if they are linked to patients phones, and which networks can be included, would be additionally helpful.

Line 198: Please include more information on who the social supporters were (e.g. inclusion/exclusion criteria). If they were required for all participants (and how many might not have met this criteria) and what their roles were given that they were ‘not given specific instructions’ (authors only state they explained expectations, but no what those were). See disclosure issue below as well.

Line 231: Loss to follow-up not included, therefore per-protocol study (vs ITT).

Line 236: ITT secondary analysis (it might be clearer to describe PP vs ITT upfront as its mentioned in the discussion)

Line 281: What is meant by social supporters’ co-infection with HIV?

Table 1: Given the question was asked of PLTB if they had ‘enough social support’, it would be interesting given the nature of the intervention to describe the experiences of the subgroup who initially reported ‘not enough social support’ at baseline and if there was any finding worth exploring in future innovations or studies.

Disclosed TB status – please explain how can 14-20% not have disclosed to anyone other than a healthcare provider be but all identified social supporters to receive texts?

Line 291: All found SMS easy to use. Is this selection bias? What is the baseline literacy and texting rate in the clinic population?

Line 295: States SMS reminders occurred when the electronic adherence monitor was opened. But isn’t the point of a reminder to occur before medication is taken? Sentence may need clarification.

Line 298: Yes, since the SMS text are medication ‘reminders’ one would expect daily reminders timed with dosing schedules. This was confirmed by patients in line 302. Note, HIV studies that use weekly texts were not reminders, they were checkins, or informational messages. The sentence in line 298 has a wording issue stating “SMS reminders to triggered SMS reminders” What does this mean?

Interesting finding about preferences of personalized messages to default ‘reminders’.

Table 2

As above, if only 80% of medication reminders are sent/received as expected, this means they would not be feasible/reliable as ‘dose reminders’ as this is below desired adherence levels? (albeit they may have other general benefits?)

SMS notifications and social supporters – this is the most novel and fascinating part of the study and is well described. The financial and social support delineation is very interesting.

Real-time adherence monitor – patient acceptability and ease of use is encouraging. However, data on the clinic population as a whole would be important as inclusion criteria may have excluded less literate and technically capable patients.

Table 3 is well presented and contains useful technical and contextual information for the readers.

Discussion of risks:

Since patients reported usefulness of the SMS to remind them to take their daily medications, discuss the risk of the 20% delayed or not delivered reminders. Are there alternatives, or does the data indicated it doesn’t matter as good habits are formed?

Table 4 has excellent illustrative examples of quotations for context.

Primary adherence data is well presented.

Discussion

The discussion is balanced and reflects the key findings. Perhaps accessibility/acceptability is overstated if intended to reflect the whole PLTB clinic population. What about the others, how would they be treated or supported in such a program?

A larger trial is recommended. Which features would the authors recommend? It seems to me the weekly ‘reminders’ arm could be discontinued due to lower effect and patient preference compared to daily. This would be a reasonable conclusion to include.

The involvement and experience of social supporters via SMS is novel and interesting.

Line 431 states this technology can keep PLTB connected with healthcare workers, but it wasn’t clear from this study how that was the case (vs social supporters).

7. PLOS authors have the option to publish the peer review history of their article (what does this mean?). If published, this will include your full peer review and any attached files.

**Do you want your identity to be public for this peer review?** For information about this choice, including consent withdrawal, please see our Privacy Policy.

Reviewer #1: No

Reviewer #2: **Yes: **Haron Gichuhi

Reviewer #4: No

Reviewer #5: No

---

## [Decision Letter · Decision Letter 3]

8 Nov 2023

The Feasibility, Acceptability, and Preliminary Impact of Real-Time Monitors and SMS on Tuberculosis Medication Adherence in Southwestern Uganda: Findings from a Mixed Methods Pilot Randomized Controlled Trial

PGPH-D-23-00372R3

Dear Dr Musiimenta

We are pleased to inform you that your manuscript 'The Feasibility, Acceptability, and Preliminary Impact of Real-Time Monitors and SMS on Tuberculosis Medication Adherence in Southwestern Uganda: Findings from a Mixed Methods Pilot Randomized Controlled Trial' has been provisionally accepted for publication in PLOS Global Public Health.

Best regards,

Megan Coffee, MD, PhD

Academic Editor

Reviewer Comments (if any, and for reference):

Reviewer's Responses to Questions

**Comments to the Author**

1. If the authors have adequately addressed your comments raised in a previous round of review and you feel that this manuscript is now acceptable for publication, you may indicate that here to bypass the “Comments to the Author” section, enter your conflict of interest statement in the “Confidential to Editor” section, and submit your "Accept" recommendation.

Reviewer #4: All comments have been addressed

Reviewer #5: All comments have been addressed

2. Does this manuscript meet PLOS Global Public Health’s publication criteria? Is the manuscript technically sound, and do the data support the conclusions? The manuscript must describe methodologically and ethically rigorous research with conclusions that are appropriately drawn based on the data presented.

Reviewer #4: Yes

Reviewer #5: Yes

3. Has the statistical analysis been performed appropriately and rigorously?

Reviewer #4: Yes

Reviewer #5: Yes

4. Have the authors made all data underlying the findings in their manuscript fully available (please refer to the Data Availability Statement at the start of the manuscript PDF file)?

Reviewer #4: Yes

Reviewer #5: Yes

5. Is the manuscript presented in an intelligible fashion and written in standard English?

Reviewer #4: Yes

Reviewer #5: Yes

6. Review Comments to the Author

Reviewer #4: (No Response)

Reviewer #5: My comments have been addressed.

7. PLOS authors have the option to publish the peer review history of their article (what does this mean?). If published, this will include your full peer review and any attached files.

**Do you want your identity to be public for this peer review?** For information about this choice, including consent withdrawal, please see our Privacy Policy.

Reviewer #4: No

Reviewer #5: **Yes: **Richard Lester
